# Antimicrobial Resistance and Genetic Characterization of *Streptococcus equi* subsp. *zooepidemicus* in Equines from Central Italy: Insights from a One Health Perspective

**DOI:** 10.3390/ani15182713

**Published:** 2025-09-16

**Authors:** Alessandra Alessiani, Marina Baffoni, Daniela Averaimo, Maria Chiara Cantelmi, Antonio Coccaro, Marco Rulli, Vanessa Piersanti, Cinzia Pompilii, Francesca Cito, Alexandra Chiaverini, Antonio Petrini

**Affiliations:** 1Istituto Zooprofilattico Sperimentale dell’Abruzzo e del Molise “G. Caporale”, 64100 Teramo, Italy; d.averaimo@izs.it (D.A.); m.cantelmi@izs.it (M.C.C.); antoniococcaro00@gmail.com (A.C.); m.rulli@izs.it (M.R.); vanessa.piersanti212@gmail.com (V.P.); c.pompilii@izs.it (C.P.); f.cito@izs.it (F.C.); a.chiaverini@izs.it (A.C.); a.petrini@izs.it (A.P.); 2Dipartimento di Medicina Clinica, Sanità Pubblica, Scienze della Vita e dell’Ambiente, Università degli Studi dell’Aquila, 67100 L’Aquila, Italy

**Keywords:** *Streptococcus equi* subsp. *zooepidemicus*, equine infection, antimicrobial resistance, One Health, Italy

## Abstract

*Streptococcus equi* subsp. *zooepidemicus* (SEZ) is a bacterium commonly found in horses but can also infect other animals and humans. In 2021–2022, Abruzzo experienced two SEZ outbreaks: one in humans linked to unpasteurized cheese (37 cases) and another in donkeys (4 deaths). This led researchers to study SEZ in horses, donkeys, and a mule in Abruzzo and Molise. Sixty-one nasal and genital swabs were tested for antibiotic resistance and genetic traits. Results showed that 37.7% of SEZ strains were resistant to at least one antibiotic, with tetracycline and trimethoprim/sulfamethoxazole being the most common. About 8.2% of strains were multidrug-resistant (MDR). Genetic testing identified resistance genes like *tet*(*W*) and *erm*(*B*), and virulence genes like *Fbp54* (which helps SEZ attach to host tissues). This study found no major clusters of related SEZ strains but detected ST61, linked to the human outbreak. The presence of antibiotic-resistant SEZ, including strains with multiple resistance genes, underscores the importance of meticulous monitoring to prevent the spread of resistant bacteria, particularly under a One Health approach that considers human, animal, and environmental health in tandem.

## 1. Introduction

*Streptococcus equi* subsp. *zooepidemicus* (SEZ) is generally regarded as a commensal organism or an opportunistic pathogen of the equine respiratory tract, oral cavity, pharynx, endometrium, and other mucosal sites. It is a Lancefield group C β-hemolytic streptococcus. Occasionally, it has been recognized as a pathogen for swine, birds, dogs, felines, other animals, and humans [1,2,3,4,5,6,7]. SEZ can cause severe infections in humans, including sepsis, septic arthritis, meningitis, and pneumonia [8,9]. An example of the power of SEZ outbreaks is the Icelandic epidemic respiratory disease involving a population of 77,000 horses [9]. In Italy, during 2021–2022, two significant SEZ-related events occurred in the Abruzzo region. The first was a human outbreak, linked to the consumption of cheese made from unpasteurized milk, resulting in 37 cases over a seven-month period, 19 of which required hospitalization. This outbreak was associated with sequence type (ST) 61 [10]. The second event involved the death of four donkeys and was attributed to a novel sequence type, ST525, which was characterized by the presence of the *mf2* virulence gene [2]. In response to its involvement in the identification and management of both outbreaks, the Istituto Zooprofilattico Sperimentale dell’Abruzzo e del Molise (IZSAM) initiated a study to characterize the SEZ population in equids within its regions of competence [11].

Equids were selected not only for their symbolic relevance in the historical context of SEZ but also for their significant role within the One Health framework. These animals influence zoonotic disease dynamics, contribute to therapeutic and rehabilitative practices, impact environmental health, and support economic resilience. Their close interactions with humans, other animals, and shared environments create potential pathways for the transmission of antimicrobial-resistant (AMR) pathogens [9,12,13]. In recent work from the University of Pisa, Bacci et al. (2024) describe antimicrobial prescribing patterns in horses over 11 years (2011–2021), showing that antimicrobials were frequently used for surgical prophylaxis and in approximately one-third of clinical cases [14]. The most commonly prescribed antibiotics were gentamicin, benzylpenicillin, and ceftiofur, with penicillin, cephalosporins, and aminoglycosides being widely used. WHO High-Priority Critically Important Antimicrobials accounted for a significant proportion of prescriptions [14]. This highlights the urgent need to address AMR through a comprehensive and integrated One Health approach [5,15].

Particular emphasis was placed on the surveillance of antibiotic resistance, a key factor in bacterial adaptation to environmental pressures and in the development of persistence and virulence traits. A total of 61 nasal and genital swabs, positive for SEZ, were collected from 490 equids in the Abruzzo and Molise regions. Regarding SEZ strains, both antimicrobial susceptibility testing and genomic characterization were conducted to identify antimicrobial resistance determinants. Among the isolates, 62.3% of strains were susceptible to all tested antimicrobials while 37.7% showed resistance to at least one. Of the resistant strains, 47.83% exhibited resistance to tetracycline and 43.48% to trimethoprim/sulfamethoxazole. Multidrug resistance (MDR) was observed in 8.20% of the isolates. The most frequently detected resistance gene was *tet*, with sporadic detection of *erm*(*B*), *ant*(*6*)*-Ia*, *cat*(*pC194*), *mef*(*2*) and *mef*(*3*), and *lsa*(*C*). Identified virulence genes included *Fbp54*, *mf2*, *mf3*, *speI*, *speK*, and *hasC*.

## 2. Materials and Methods

### 2.1. Samples

Sixty-one *Streptococcus equi* subspecies *zooepidemicus* (SEZ) strains were selected between 2022 and 2024. The strains were isolated by analyzing 490 equids and obtained from 49 nasal and urogenital swab of horses, donkeys, and a mule, located in the Abruzzo and Molise regions. SEZ detection was carried out in a CO_2_-enriched atmosphere, at 37 °C, on Edwards modified medium (Microbiol, Cagliari, Italy). The description of the selected horses was developed by referring by Cito et al. (2025) [11], and samples from donkeys and the mule were taken in the laboratory by company veterinarians for diagnostic purposes.

Identification was performed by MALDI-TOF-MS spectrometer and MALDI Biotyper (Bruker Daltonics, Bremen, Germany) analysis, according to the manufacturer’s instructions. Isolated strains were subjected to an antibiogram, as indicated in Section 2.2, and DNA extraction was carried out as indicated in Section 2.3. When DNA extraction could not be performed immediately, the strains were stored in a Microbank™ (Biolife italiana s.r.l., Milano, Italy) at −20 °C.

### 2.2. Antimicrobial Susceptibility Test

SEZ strains’ antimicrobial susceptibility was characterized phenotypically by broth microdilution using Sensititre^®^ GPALL1F^®^ plates and the Sensititre^®^ SWIN Software System^®^ 3.4 (Termofisher Scientific, Paisley, UK). GPALL1F^®^ plates contain chloramphenicol (CHL, 2–16 µg/mL), erythromycin (ERY, 0.25–4 µg/mL), clindamycin (CLI, 0.5–2 µg/mL), daptomycin (DAP, 0.5–4 µg/mL), oxacillin NaCl2% (OXA+, 0.25–4 µg/mL), streptomycin 1000 (STR 1000 µg/mL), gentamicin (GEN, 2–16 µg/mL), ampicillin (AMP, 0.12–8 µg/mL), linezolid (LZD, 1–8 µg/mL), penicillin (PEN, 0.06–8 µg/mL), vancomycin (VAN, 0.25–32 µg/mL), trimethoprim/sulfamethoxazole (SXT, 0.5/9–4/76 µg/mL), levofloxacin (LEVO, 0.25–4 µg/mL), ciprofloxacin (CIP, 1–2 µg/mL), quinupristin/dalfopristin (SYN, 0.5–4 µg/mL), tigecycline (TGC, 0.03–0.5 µg/mL), nitrofurantoin (NIT, 32–64 µg/mL), tetracycline (TET, 2–16 µg/mL), moxifloxacin (MXF, 0.25–4 µg/mL), gentamicin 500 µg/mL (GEN 500 µg/mL), and rifampin (RIF, 0.5–4 µg/mL).

Strains were classified as susceptible (S), intermediate (I), or resistant (R) according to the CLSI (Clinical and Laboratory Standard Institute) VET01S ED7:2024 when there were specific indications for horses; otherwise, CLSI M100-ED34:2024 was used for non-horses’ specific indications (for levofloxacin, linezolid, quinupristin/dalfopristin) [16,17] or EUCAST Clinical breakpoint v. 14.0 2024 (for daptomycin, rifampin, and tigecycline) [18]. Antimicrobials lacking species-specific breakpoints for *Streptococcus* spp. were excluded (streptomycin 1000, gentamicin, oxacillin NaCl2%, and ciprofloxacin).

A multidrug-resistant (MDR) strain is defined as non-susceptible to at least one agent in each of three or more antimicrobial categories [19].

### 2.3. Whole-Genome Sequence Analysis, MLST, Virulence Factor, and Antimicrobial Resistance Genetic Profile

DNA extraction was performed according to Portman et al. (2018) [20] with minor modifications, and next-generation sequencing (NGS) was performed through the Illumina platform (Illumina, San Diego, CA, USA). For the WGS data analysis, an in-house pipeline (https://github.com/genpat-it/ngsmanager/ accessed on 12 December 2024) was used, and a quality check was performed. To confirm species identification, the KmerFinder v. 3.0.2 tool was used [21]. Multilocus Sequence Typing (MLST) was performed according to the reference scheme (https://pubmlst.org/organisms/streptococcus-zooepidemicus, accessed on 12 December 2024). ResFinder v.4.7.2 [22] was used to detect antimicrobial resistance genes using default parameters.

## 3. Results

### 3.1. Samples

In total, 61 samples tested positive for SEZ: 49 nasal swabs and 12 genital swabs. Forty-nine SEZ-positive swabs were collected from horses, eleven from donkeys, and one from a mule, as shown in Table 1.

### 3.2. Antimicrobial Susceptibility Test

Thirty-eight (62.3%) strains were susceptible to all antimicrobials tested, and twenty-three (37.7%) were resistant to one or more one antimicrobials, as shown in Figure 1.

Eleven (47.83% of the resistant) strains were resistant to tetracycline, ten (43.48%) to trimethoprim/sulfamethoxazole, eight (34.78%) to daptomycin, seven (30.43%) to moxifloxacin, six (26.09%) to clindamycin and vancomycin, four (17.39%) to erythromycin and tigecycline, and three (13.04%) to chloramphenicol, ampicillin, linezolid, quinupristin/dalfopristin, and penicillin. As shown in Figure 2, antimicrobials that were excluded due to a lack of reference or a dilution range outside clinically relevant thresholds were oxacillin, nitrofurantoin, levofloxacin, streptomycin, ciprofloxacin, gentamycin, and rifampin.

Five (21.74%) strains were identified as MDR. One strain (identified as ID 20 in Table 2), isolated from a horse genital swab, exhibited MDR 4 resistance to DAP, TET, SXT, and VAN. Another strain (ID 22), from a horse nasal swab, displayed MDR 5 resistance to DAP/CLI, MXF, TGC, SXT, and VAN. Two strains from horse nasal swabs showed MDR 7 profiles: one (ID 15) resistant to AMP/PEN, CHL, CLI/ERY, LZD, SYN, TET/TGC, and VAN, and the other (ID 47) resistant to AMP/PEN, CHL, DAP/CLI/ERY, LZD, SYN, TET/TGC, and VAN. Additionally, a strain from a mule nasal swab (ID 48) demonstrated an MDR 9 pattern, with resistance to AMP/PEN, CHL, DAP/CLI/ERY, LZD, MXF, SYN, TET/TGC, SXT, and VAN. Table 2 provides detailed genotype and phenotype data for selected MDR strains, tetracycline-resistant strains, and MLSB (macrolide–lincosamide–streptogramin B) strains.

### 3.3. Whole-Genome Sequence Analysis, MLST, Virulence Factor, and Antimicrobial Resistance Genetic Profile

MLST detected 39 different STs of 61 strains: 5 (8.20%) strains for ST 71, 4 (6.56%) strains each for ST97, ST332, and ST 550, 3 (4.91%) strains each for ST474 and ST542, 2 (3.28%) strains each for ST10, ST61, ST175, ST197, and ST541, and 1 (1.64%) strain each for ST66, ST72, ST76, ST131, ST147, ST174, ST200, ST218, ST364, ST367, ST369, ST393, ST394, ST470, ST 488, ST536, ST537, ST538, ST539, ST540, ST543, ST544, ST546, ST547, ST548, ST551, ST552, and one unknown. Virulence-detected genes were *Fbp54* on 43 strains (70.49%), *mf3* on 29 strains (47.54%), *mf2* on 23 strains (37.71%), *hasC* on 16 strains (26.23%), *speI* on 2 strains (3.28%), and *speK* on 1 strain (1.64%). One strain (identified as ID 10 in Table 2), isolated from a horse nasal swab, carried the *erm*(*B*) and *ant*(*6*)*-Ia* resistance genes. Another strain (ID 48), obtained from a mule nasal swab, harbored the *cat*(*pC194*) and *mefA* genes and was classified as MDR 9. Two strains from horse nasal swabs (ID 15 and 47) carried the *Isa*(*C*) gene, with both classified as MDR 7. Additionally, three strains (two from horse nasal swabs and one from a horse genital swab, ID 54, 59, and 60) carried the *tet*(*W*) gene. Among other findings, one strain (ID 41) from a horse nasal swab carried *tet*(*O*), while a strain from a horse genital swab (ID 36) carried *tet*(*32*). Strains harboring *tet* genes were phenotypically resistant to tetracycline.

## 4. Discussion

This work originates from the selection of equids used by Cito et al. (2025), based on the regional equine population registered in 2023 [11], and from samples taken by company veterinarians during routine activities. This makes the data shown, relating to the regional equine population, particularly significant. Although prevalence data and the sampling plan are not the subject of this article, they are available in the article by Cito et al. (2025).

MLST has become the most common way of indicating whether or not strains are related and form clusters. In the present study, no specific clusters were identified, and some STs previously reported by Nocera et al. (2023) were detected [4]. Notably, the presence of ST61 is of considerable importance, as it is the same sequence type that caused infections in humans in the Abruzzo region between 2021 and 2022 [10]. In this study, ST61 was isolated in two horses in the Molise region, but there were no detected links to human samples from the outbreak covered by Bosica et al. (2023) [10]. However, this simply means that a potentially dangerous ST is still circulating in the region. Conversely ST525, which was responsible for an outbreak in donkeys during the same period in Abruzzo [2], was not detected in the equids examined.

The virulence genes *Fbp54*, *mf3*, *mf2*, *hasC*, *speI*, and *speK* were more commonly found in *Streptococcus equi* subsp. *zooepidemicus* (SEZ) than in *Streptococcus pyogenes* (group A streptococcus, GAS), a well-known and significant human pathogen [23,24]. *Fbp54* was the most common gene detected in this study; it is associated with the production of fibronectin (Fn), which is involved in the attachment of Gram-positive cocci to host tissue. The main function of Fn is to mediate the substrate adhesion of eukaryotic cells, which involves the binding of specific cell surface receptors to certain domains of the fibronectin molecule. The protein also interacts with several other macromolecules, including DNA, heparin, fibrin, and collagen; *mf2* and *mf3* are a streptococcal superantigens, distinct from streptococcal pyrogenic exotoxins [23,25]. Although these *mf* genes were originally attributed specifically to group A streptococci, it is now known that they are commonly detected in group C streptococci as well [26,27]. The *hasC* gene is related to hyaluronic acid synthesis. SEZ’s characteristic ability to produce hyaluronic acid is useful in evading the immune system but has been heavily exploited by the cosmetics industry, which considers group C streptococci to be non-pathogenic to humans [25]. In this study, *hasC* was frequently found to be associated with mitogenic factors and *Fbp54*—in one case, in an MDR7 strain, and in another case, also with *speI* and *speK. speI* and *speK* are superantigen genes capable of stimulating the immune system to release inflammatory cytokines; they are homologous to superantigens in GAS [23,25]. In this study, one strain isolated from a horse nasal swab was found to harbor the *Fbp54*, *mf2*, *speI*, *speK*, and *hasC* genes. Although no genotypic or phenotypic elements of antibiotic resistance were detected, this genotype is extremely peculiar and potentially dangerous for both horses and humans.

Another relevant aspect in the study of the pathogenicity and adaptability of *Streptococcus equi* subsp. *zooepidemicus* (SEZ) concerns the presence of antibiotic resistance genes such as *cat*(*pC194*), *mef2*, and *mef3*. The *cat*(*pC194*) gene, originally identified on streptococcal plasmids, encodes a chloramphenicol acetyltransferase that inactivates the drug by acetylation, conferring resistance to chloramphenicol [28]. Although this antibiotic is now rarely used in clinical practice, the persistence of this gene remains an indicator of the bacterium’s ability to acquire and maintain resistance determinants through mobile genetic elements. The *mef2* and *mef3* genes, on the other hand, encode efflux pumps belonging to the Major Facilitator Superfamily (MFS) and are associated with the so-called M-resistance phenotype, which actively expels macrolides from the bacterial cell [29]. In SEZ, these genes have been reported in strains isolated from both animal and human sources, highlighting the zoonotic nature and adaptability of the pathogen [30]. The coexistence of virulence determinants (superantigens, adhesion factors, genes for hyaluronic acid synthesis) and antimicrobial resistance genes strengthens the hypothesis that *S. zooepidemicus* is not only an opportunistic pathogen of equids but also a potential emerging threat to public health, especially in contexts of close human–animal contact.

Among the total number of samples examined, 37.7% were found to be resistant to at least one antibiotic. This is a rather unusual finding compared to other studies on SEZ, published over the years. In fact, authors closer in time and space, such as Nocera et al. (2021–2023) or Burgio et al. (2024), reported that all the strains they studied exhibited at least one resistance [4,31,32]. The difference between the works of the cited authors and this one is probably due to the detection method used (disk diffusion versus broth microdilution) and since the studies were conducted in different areas, with different animal management practices, and at different times.

The analysis revealed that tetracycline resistance was the most frequently observed trait among the strains, with 11 strains (47.83% of the resistant isolates) exhibiting resistance. However, only five of these strains carried identifiable resistance genes: three carried *tet*(*W*), one carried *tet*(*O*), and one carried *tet*(*32*). The *tet*(*W*) gene, recently identified in streptococci, encodes for a protective enzyme, while *tet*(*O*) encodes ribosome-protective proteins [33]. The mosaic gene *tet*(*32*), first detected in *Streptococcus suis*, integrates elements from *tet*(*O/W/32/O*) and is frequently associated with integrative and conjugative elements (ICE), facilitating rapid horizontal gene transfer among pathogenic streptococci [34,35]. Despite this, six strains showed phenotypic resistance to tetracycline in the absence of detectable *tet* genes. In comparison, some previous studies refer to different resistance rates: the EFSA reports 29.3–47.5% in 2021 [36], Nocera et al. report 86.4% in 2023 [4], Kabir et al. report 85.3% in 2024 [5], Isgren et al. (2020) report 33.8% [12], and Leon et al. (2020) report almost 60% [13]. The variable results reported by different authors over time are due not only to spatial and temporal distances but also to the well-documented fact that rates of antimicrobial resistance measured by broth microdilution (BMD) are often lower than those obtained by disk diffusion (DD) [37,38]. A combination of methodological, medium-related, and biological factors influences the lower resistance rates observed by BMD compared to DD [37,38,39]. Other recent studies show the presence of transmissible resistance genes, along with concerning epidemiological trends, which suggest a potential increase in tetracycline resistance within the studied regions. The second most common resistance observed was to sulfamethoxazole/trimethoprim, affecting 43.48% of the resistant strains. This resistance is typically associated with plasmids carrying *sul* genes. Notably, resistance to sulphonamides emerged shortly after their introduction into clinical practice [40]. Given its proven efficacy in previous studies and the rarity of adverse events, it is indeed possible that the product was used on equids [41]. However, no *sul* genes were detected in the examined strains. Resistance to macrolides, lincosamides, and streptogramin B antibiotics (MLS group), such as clindamycin, erythromycin, daptomycin, and quinupristin/dalfopristin in this study, was observed. These antibiotics target bacterial protein synthesis, and resistance is commonly mediated by *erm*, *Isa*(*C*), and *mef*(*A*) genes [40]. Three strains carried genes linked to MLS resistance, but only two displayed a full MLS phenotype. One strain was resistant to macrolides and lincosamides but remained sensitive to streptogramins. Among all resistant strains, resistance rates to individual MLS antibiotics were 34.78% for daptomycin, 26.09% for clindamycin, and 17.39% for erythromycin. Only 13.04% of strains harbored MLS resistance genes, which may have been lost during sample storage or revitalization. A strain isolated from a horse nasal swab carried both *ant*(*6*)*-Ia* and *erm*(*B*) genes but did not exhibit an MDR or complete MLSB resistance phenotype. The *ant*(*6*)-Ia gene enables resistance to aminoglycosides through nucleotidyltransferase-mediated modification, while *erm*(*B*) encodes an rRNA methylase, responsible for MLSB resistance [33,40]. Despite harboring these genes, the strain remained susceptible to streptogramins and showed resistance only to sulfamethoxazole/trimethoprim. The low prevalence of strains phenotypically resistant to penicillin (13.04%) may be attributed to SEZ’s ability to express phenotypically persistent cells, as demonstrated in vitro. Nevertheless, this rate is significantly lower than that reported by Pisello et al. (2019) in a study conducted in central Italy [42]. Among the MDR strains, MDR 9 harbored the *cat*(*pC194*) gene, responsible for chloramphenicol resistance via acetyltransferase activity, as well as the *mef*(*A*) gene, which mediates the macrolide efflux pump [33,40]. These genes are frequently located on conjugative transposons like Tn1207.3, contributing to MLSB resistance [33]. *mef*(*A*) is an acquired gene involved in the MLSB-resistant phenotype; this group also includes linezolid with same action mechanism, which explains the reason for such extensive resistance. However, no specific resistance genes have been detected for tetracycline or beta lactams, nor for trimethoprim/sulfamethoxazole and glycopeptides. Among the MDR strains, seven carried the *Isa*(*C*) gene, which confers MLSB resistance via efflux pumps, but no resistance genes were detected for tetracycline, beta-lactams, or glycopeptides, despite phenotypic resistance. In contrast, MDR 4 and MDR 5 strains did not harbor any detectable resistance genes.

In similar studies conducted in Italy in recent years, no significant phenotypic similarities were observed. For instance, the study carried by Nocera et al. (2024) [4] showed resistance rates considerably higher than those in the present study: tetracycline resistance was 86.4% in Nocera et al. versus 47.83% in this study, and resistance to sulfamethoxazole/trimethoprim was 95.4% versus 43.48%, respectively. Similarly, in the study by Burgio et al. (2024) [28], all tested strains were resistant to sulfamethoxazole/trimethoprim, whereas in the present study, only 10 out of 61 strains exhibited such resistance. Even in comparison with international studies, the resistance rates observed in our region appear lower. For example, Kabir et al. (2024) [5] reported that 99.6% of *Streptococcus equi* subsp. *zooepidemicus* (SEZ) strains examined in the USA were resistant to at least one antibiotic, with 85.3% resistant to tetracycline. Furthermore, Kabir et al. noted that 53.3% of the strains were MDR, compared to 21.74% in our study [5].

From a molecular perspective, the findings also appear to be highly variable. The only consistent feature observed across studies is the presence of the *Isa*(*C*) gene, which was identified both in our study and in the work by Nocera et al. (2024) [4]. Notably, Nocera et al. (2024) also reported frequent discrepancies between phenotypic resistance profiles and those predicted in silico using molecular methods [4]. The discrepancy between phenotypic antimicrobial susceptibility testing and genomic predictions obtained through next-generation sequencing is a well-documented phenomenon that reflects the complexity of antimicrobial resistance mechanisms. Although genomic approaches allow for the detection of a wide range of known resistance determinants, they may overlook resistant subpopulations that occur at low levels but can still have a significant impact on phenotypic assays [43]. Similarly, resistance genes located on plasmids or other mobile genetic elements may be underrepresented or lost during DNA extraction and assembly. This results in their absence from genomic data, despite their contribution to resistance phenotypes [44].

## 5. Conclusions

This study provides novel insights into *Streptococcus equi* subsp. *zooepidemicus* in Italian territories. Integrating the analysis of minimum inhibitory concentrations with the genetic structure of the examined population and its distribution across a defined geographical area and timeframe provides valuable insights into the local epidemiological landscape. Although the overall antimicrobial resistance rates were lower than those reported in previous studies, tetracycline and sulfonamide–trimethoprim resistance remained the most prevalent traits. The presence of multidrug-resistant strains carrying genes such as *cat*(*pC194*), *mef*(*A*), and *Isa*(*C*) highlights the adaptability of SEZ and its ability to maintain mobile resistance elements. This study shows that microdilution-based methods result in lower overall resistance rates than disk diffusion methods, emphasizing the need for large-scale standardization of susceptibility testing techniques. Potential discrepancies between phenotypic and genotypic resistance profiles, as well as possible links between colony morphology and underlying genotypic variations, warrant further investigation. From a One Health perspective, the detection of sequence types previously associated with human infections, such as ST61, which was identified in Abruzzo between 2021 and 2022, highlights the zoonotic potential of SEZ. The presence of virulence determinants (superantigens, adhesion factors, and hyaluronic acid synthesis genes) alongside antimicrobial resistance genes suggests an escalating pathogenic potential, similar to that of Streptococcus pyogenes (group A streptococcus). These findings highlight the importance of implementing systematic and coordinated surveillance that integrates veterinary and human health monitoring. This approach is crucial for identifying emerging high-risk clones and limiting cross-species transmission. Controlling SEZ infections in equids is therefore critical for public health, particularly in contexts of close human–equid interaction.

## Figures and Tables

**Figure 1 animals-15-02713-f001:**
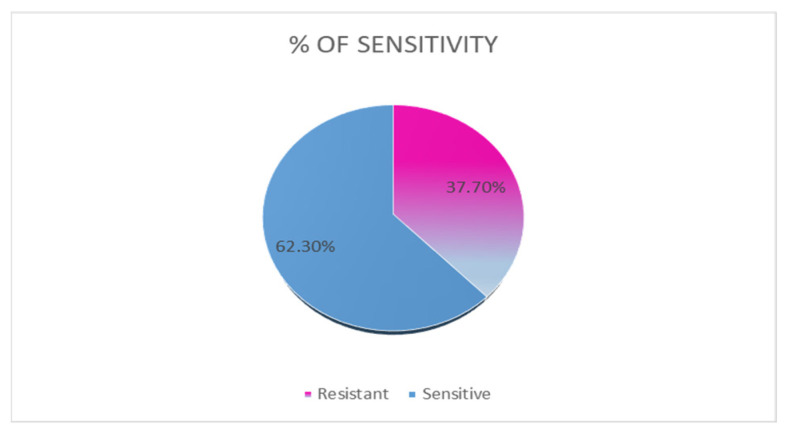
Percentage of SEZ strains with susceptible or resistant phenotypes to tested antimicrobials.

**Figure 2 animals-15-02713-f002:**
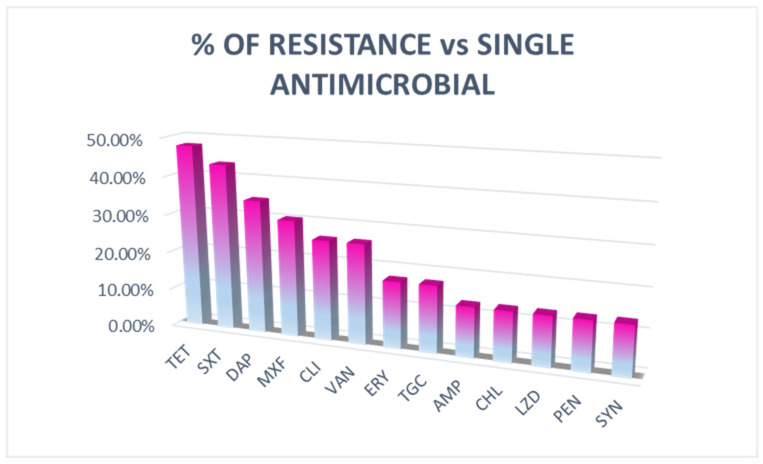
Percentage of antimicrobial resistance on single antimicrobial. Tetracycline (TET), trimethoprim/sulfamethoxazole (SXT), daptomycin (DAP), moxifloxacin (MOX), clindamycin (CLI), vancomycin (VAN), erythromycin (ERY), tigecycline (TGC), ampicillin (AMP), linezolid (LZD), penicillin (PEN), and quinupristin/dalfopristin (SYN).

**Table 1 animals-15-02713-t001:** Sample types that are SEZ-positive.

Species	Nasal Swabs	Genital Swabs	Total
HORSE	37	12	49
DONKEY	11	0	11
MULE	1	0	1
TOTAL	49	12	61

**Table 2 animals-15-02713-t002:** Genotypes and phenotypes of some resistant strains.

ID	Animal Swabs		Resistance Genes				STs	Virulence Genes	Phenotype Details
10	HORSE	NASAL	*ant*(*6*)*-Ia*	*erm*(*B*)			536	*mf3*	DAP/ERY-SXT
15	HORSE	NASAL		*lsa*(*C*)			147	*Fbp54*, *mf3*	AMP/PEN-CHL-DAP/CLI/ERY-LZD-SYN-TET/TGC-VAN
47	HORSE	NASAL		*lsa*(*C*)			547	*Fbp54*, *mf3*, *hasC*	AMP/PEN-CHL-CLI/ERY-LZD-SYN-TET/TGC-VAN
48	MULE	NASAL		*mef*(*A*)	*cat*(*pC194*)		369	*Fbp54*, *mf3*	AMP/PEN-CHL-DAP/CLI/ERY-LZD-MXF-SYN-TET/TGC-SXT-VAN
36	HORSE	GENITAL				*tet*(*32*)	364	*mf3*	MOX-TET
41	HORSE	NASAL				*tet*(*O*)	470	*mf3*	CLI-TET
54	HORSE	NASAL				*tet*(*W*)	61	*Fbp54*, *mf2*, *hasC*	TET
59	HORSE	NASAL				*tet*(*W*)	61	*Fbp54*, *mf2*	TET
60	HORSE	GENITAL				*tet*(*W*)	72	*Fbp54*, *mf2*	TET

## Data Availability

The original contributions presented in this study are included in the article. Further inquiries can be directed to the corresponding authors.

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
