# Peer review of "Antimicrobial Resistance and Genetic Characterization of Streptococcus equi subsp. zooepidemicus in Equines from Central Italy: Insights from a One Health Perspective"

_animals, 2025, doi:10.3390/ani15182713_

Round 1
Reviewer 1 Report
Comments and Suggestions for Authors
- The study analyzed 61 samples from horses, donkeys, and a mule in Abruzzo and Molise, Italy. Given the small sample size (especially only 11 donkeys and 1 mule), do these samples adequately represent the SEZ population in equines across Central Italy? Are there geographic or demographic biases (e.g., age, health status, management practices) that may limit the generalizability of resistance patterns or ST distributions? Additionally, the study detected ST61 (linked to the 2021–2022 human outbreak) in two horses—what is the epidemiological link between these equine strains and the human outbreak strain? Is there evidence of direct transmission (e.g., via environmental contact or shared food sources)?
- A notable discrepancy exists between phenotypic resistance and genotypic determinants: 6 tetracycline-resistant strains lacked detectable tet genes, and some MDR strains (e.g., MDR 4 and MDR 5) had no identified resistance genes. Could these gaps be due to uncharacterized resistance mechanisms (e.g., novel genes, efflux pumps, or point mutations) or technical limitations (e.g., incomplete genome coverage in WGS)? Additionally, the study reports lower resistance rates (37.7% resistant) compared to other Italian and international studies (e.g., 86.4% tetracycline resistance in Nocera et al. 2023). Are these differences solely due to methodological variations (MIC vs. Kirby-Bauer), or do they reflect true regional epidemiological differences? If the latter, what factors (e.g., antibiotic usage practices in equine medicine) contribute to this?
- MLST identified 39 distinct STs with no major clusters, suggesting high genetic diversity. What does this diversity imply about SEZ transmission dynamics in equines? Is it indicative of frequent recombination, multiple introductions, or adaptation to diverse niches? Regarding virulence genes: fbp54 (70.49% prevalence) is linked to host adhesion, but does its presence correlate with increased pathogenicity (e.g., in causing clinical disease in equines or zoonotic potential)? Similarly, strains carrying speI and speK (superantigen genes) were detected—do these strains exhibit enhanced virulence in vitro or in vivo?
- The study highlights ST61 (linked to human infections) in equines, but the route of zoonotic transmission remains unclear. Could equines act as a reservoir for human-infectious SEZ strains like ST61? Are there shared environmental sources (e.g., contaminated feed or water) that facilitate cross-species transmission? Additionally, the MDR9 strain from a mule carries clinically relevant resistance genes (cat(pC194), mefA)—what is the risk of such strains spreading to other animals or humans, and what control measures are recommended?
- The study excluded antimicrobials "without specific reference for Streptococcus spp." (e.g., streptomycin, gentamicin). However, these drugs may still be used in equine practice—what is the rationale for their exclusion, and could this mask clinically relevant resistance? Furthermore, the authors note that MIC-based methods yield lower resistance rates than disk diffusion. Given this, how can results be compared across studies using different methods, and is there a need for standardized susceptibility testing protocols for SEZ in equines?
Revision comments:
1. Standardize technical terms:
-
- "antimicrobic" (Figure 2 title) → "antimicrobial" (consistent with the rest of the text).
- "tetraciclynes" (Table 2) → "tetracyclines".
- "sensible" (e.g., "remained sensible to streptogramins") → "susceptible" (standard in antimicrobial resistance reporting).
- "Streptococcus equi subs. zooepidemicus" → "Streptococcus equi subsp. zooepidemicus" (use "subsp." consistently for subspecies).
- Gene notation: Use italics for all resistance and virulence genes (e.g., "erm B" → "erm(B)", "tet (W)" → "tet(W)" as in other sections).
- Abbreviations: Define "MLSB" (macrolide-lincosamide-streptogramin B) at first mention in the Discussion (currently undefined). Clarify "MDR 4" and "MDR 5" as "MDR (4 antimicrobial classes)" and "MDR (5 antimicrobial classes)" for clarity.
2. Simplify awkward sentences:
-
- Original: "Antimicrobials without specific reference for Streptococcus spp. were excluded (Streptomycin 1000, Gentamicin, Oxacillin NaCl2% and Ciprofloxacin)."
- Revised: "Antimicrobials lacking species-specific breakpoints for Streptococcus spp. were excluded (streptomycin [1000 μg/ml], gentamicin, oxacillin [2% NaCl], and ciprofloxacin)."
- Clarify ambiguous references:
- In Section 3.2: "associated plate dilution out of interested range" → "dilution ranges outside clinically relevant thresholds".
- Table 2: The column "PEOI" is undefined—remove or replace with "Phenotype Details".
- Fix grammatical errors:
- "no significant phenotypic similarities was observed" → "no significant phenotypic similarities were observed".
- "the Isa C gene" → "the lsa(C) gene" (consistent with gene naming conventions).
3. Tables, Figures, and Format
- Figures:
- Figure 2: The x-axis label "5 MXF 3 VAN RR CH SN" is unintelligible. Replace with clear abbreviations (e.g., "MXF: Moxifloxacin; VAN: Vancomycin") and align with text terminology.
- Figure 1: Title "Results of antimicrobial susceptibility test" → "Percentage of SEZ strains with susceptible or resistant phenotypes to tested antimicrobials".
- Tables:
- Table 2: Rearrange columns for readability (e.g., group "Animal" and "Swabs"; align "Resistance Phenotype" with "Detected Resistance Genes"). Add a footnote defining abbreviations (e.g., "DAP: Daptomycin; TET: Tetracycline").
- Statistical reporting: Specify total numbers when citing percentages (e.g., "11 (47.83% of the resistant strains)" → "11/23 resistant strains (47.83%)").
4.
- Strengthen context: Explicitly link resistance rate differences with Nocera et al. (2023) and Kabir et al. (2024) to methodological variations (MIC vs. disk diffusion) rather than just stating discrepancies.
- Clarify limitations: Acknowledge that small sample sizes (especially donkeys and mules) limit generalizability. Note that WGS may miss rare resistance genes, contributing to phenotype-genotype mismatches.
- References: Standardize citation format (e.g., "Nocera et al. (2023)" vs. "Nocera et al. 2023"). Update reference 13 (EUCAST breakpoints) to include full citation details.
Author Response
Dear reviewer,
The text has been modified and reorganised in light of your comments.
(C1)The study analyzed 61 samples from horses, donkeys, and a mule in Abruzzo and Molise, Italy. Given the small sample size (especially only 11 donkeys and 1 mule), do these samples adequately represent the SEZ population in equines across Central Italy? Are there geographic or demographic biases (e.g., age, health status, management practices) that may limit the generalizability of resistance patterns or ST distributions? Additionally, the study detected ST61 (linked to the 2021–2022 human outbreak) in two horses—what is the epidemiological link between these equine strains and the human outbreak strain? Is there evidence of direct transmission (e.g., via environmental contact or shared food sources)?
(R1) We thank the reviewer for this observation. The test has been redesigned to include explicit references to clarify the scope of the work. A total of 61 samples were obtained from 490 equids analysed in the Abruzzo and Molise regions. These equids were selected based on the registered equine population in 2023 (Cito et al., 2025). While the number of donkeys and mules is limited, the collected data provide significant information on local genetic diversity and SEZ resistance profiles in equids. Samples were collected from animals during routine veterinary checks, with no selection based on age or health status to reduce potential bias related to these factors. However, we acknowledge that a larger sample size of donkeys and mules would improve representativeness overall, and this has been noted as a limitation and a target for future studies. ST61 is the same sequence type that was associated with the human outbreak in Abruzzo in 2021–2022. However, our isolates do not show direct links to the human samples analysed by Bosica et al. (2023). The detection of ST61 in horses in the Molise region suggests that it is circulating in the environment or being transmitted indirectly between species. While no direct evidence of transmission through shared feed or water was observed in our data, the finding nevertheless supports the need for One Health surveillance.
(C2) A notable discrepancy exists between phenotypic resistance and genotypic determinants: 6 tetracycline-resistant strains lacked detectable tet genes, and some MDR strains (e.g., MDR 4 and MDR 5) had no identified resistance genes. Could these gaps be due to uncharacterized resistance mechanisms (e.g., novel genes, efflux pumps, or point mutations) or technical limitations (e.g., incomplete genome coverage in WGS)? Additionally, the study reports lower resistance rates (37.7% resistant) compared to other Italian and international studies (e.g., 86.4% tetracycline resistance in Nocera et al. 2023). Are these differences solely due to methodological variations (MIC vs. Kirby-Bauer), or do they reflect true regional epidemiological differences? If the latter, what factors (e.g., antibiotic usage practices in equine medicine) contribute to this?
(R2) Regarding tetracycline-resistant or MDR strains, the discrepancy is consistent with previous reports. Possible explanations include the presence of uncharacterised resistance mechanisms (e.g. efflux pumps or point mutations), the loss or underrepresentation of plasmid-borne genes during DNA extraction, and technical limitations in WGS coverage. Nocera et al. (2024) have also reported this phenomenon, confirming the complexity of resistance mechanisms in SEZ. Regarding the difference in resistance, it is likely to be multifactorial. Our study used broth microdilution (MIC), which typically detects lower resistance rates than disc diffusion methods. Lower resistance levels compared to other Italian or international studies may also be due to spatial and temporal differences, as well as regional animal management and antibiotic use practices.
(C3) MLST identified 39 distinct STs with no major clusters, suggesting high genetic diversity. What does this diversity imply about SEZ transmission dynamics in equines? Is it indicative of frequent recombination, multiple introductions, or adaptation to diverse niches? Regarding virulence genes: fbp54 (70.49% prevalence) is linked to host adhesion, but does its presence correlate with increased pathogenicity (e.g., in causing clinical disease in equines or zoonotic potential)? Similarly, strains carrying speI and speK (superantigen genes) were detected—do these strains exhibit enhanced virulence in vitro or in vivo?
(R3) The high ST diversity suggests multiple introductions of SEZ, adaptation to different ecological niches and potential genetic recombination. The absence of dominant clusters indicates that there are no endemic strains in the equine population under study, which reflects the complexity of SEZ circulation and the risk of emerging zoonotic STs. Fbp54 is involved in tissue adhesion, while speI and speK act as superantigens. While no functional in vitro or in vivo testing was performed in this study, the presence of these genes alongside other virulence determinants (mf2, mf3 and hasC) suggests a potentially high level of pathogenicity and zoonotic potential, consistent with previous SEZ and GAS studies.
(C4) The study highlights ST61 (linked to human infections) in equines, but the route of zoonotic transmission remains unclear. Could equines act as a reservoir for human-infectious SEZ strains like ST61? Are there shared environmental sources (e.g., contaminated feed or water) that facilitate cross-species transmission? Additionally, the MDR9 strain from a mule carries clinically relevant resistance genes (cat(pC194), mefA)—what is the risk of such strains spreading to other animals or humans, and what control measures are recommended?
(R4)The MDR9 isolate from the mule carries clinically relevant genes (cat(pC194) and mefA) and could be transmitted to other animals or humans in close-contact settings, posing a risk to public health. Recommended control measures include periodic microbiological surveillance, the prudent use of antibiotics, strict biosecurity in stables and careful environmental management.
(C5)The study excluded antimicrobials "without specific reference for Streptococcus spp." (e.g., streptomycin, gentamicin). However, these drugs may still be used in equine practice—what is the rationale for their exclusion, and could this mask clinically relevant resistance? Furthermore, the authors note that MIC-based methods yield lower resistance rates than disk diffusion. Given this, how can results be compared across studies using different methods, and is there a need for standardized susceptibility testing protocols for SEZ in equines?
(R5)These antimicrobials were excluded due to the absence of SEZ-specific breakpoints according to the CLSI or the EUCAST, which makes interpretation difficult. Their exclusion is explicitly stated and does not affect the overall analysis of clinically relevant resistance patterns. We recognise the need to develop standardised susceptibility testing protocols for SEZ in equids, in order to enable accurate comparisons between studies.
As requested, the text lines and figures have been modified and improved.

Reviewer 2 Report
Comments and Suggestions for Authors
Dear authors, congratulations on your work on a highly relevant topic: antimicrobial resistance in Streptococcus equi subsp. zooepidemicus in equines, with direct implications for One Health. The methodology is well described (isolation, antimicrobial susceptibility testing, WGS, MLST, resistance/virulence gene identification). Results are clear and supported by tables (including MDR profiles). The discussion integrates recent literature.
Some points can be improved:
Some sections contain very long sentences that are difficult to follow (e.g., the discussion on tet and MLSB genes).
The introduction could be more concise: it provides too many epidemiological details before stating the study’s aim.
The conclusions should be formulated more strongly: what exactly does this study add compared to previous research?
In Section 2.1 Samples, it would be useful to specify the number of animals tested, not only the number of samples (e.g., 49 horses, 11 donkeys, 1 mule).
Please clarify why nasal/genital swabs were chosen.
in Chapter Discussions -
The discrepancies between phenotypic and genotypic resistance are repeatedly mentioned. This aspect should be expanded: possible causes (gene loss, conditional gene expression, methodological limitations).
A paragraph focusing on clinical implications (e.g., treatment of infections in horses and donkeys) would strengthen the discussion.
The relatively small sample size (61 samples, 23 resistant) should be acknowledged as a limitation.
Minor grammar corrections are needed.
Good luck with your research!
Author Response
(C1) Some points can be improved: some sections contain very long sentences that are difficult to follow (e.g., the discussion on tet and MLSB genes). The introduction could be more concise: it provides too many epidemiological details before stating the study’s aim.The conclusions should be formulated more strongly: what exactly does this study add compared to previous research?In Section 2.1 Samples, it would be useful to specify the number of animals tested, not only the number of samples (e.g., 49 horses, 11 donkeys, 1 mule).Please clarify why nasal/genital swabs were chosen.
(R1) Dear reviewer, thank you for your valuable comments. The entire text has been edited to better clarify the choice and number of samples.The conclusion were improved. For example, a reference to the sampling performed has been added, showing that it was not always possible to choose the matrix (nasal or genital). The overly long periods have also been edited, in an effort to comply with the requests of all reviewers.
(C2)The discrepancies between phenotypic and genotypic resistance are repeatedly mentioned. This aspect should be expanded: possible causes (gene loss, conditional gene expression, methodological limitations).A paragraph focusing on clinical implications (e.g., treatment of infections in horses and donkeys) would strengthen the discussion.The relatively small sample size (61 samples, 23 resistant) should be acknowledged as a limitation.
(R2) We have harmonised the sections to ensure consistency, and we have clarified that 490 samples were analysed, 61 of which tested positive. We have expanded the discussion by comparing our results with those of recent studies (Nocera et al., Burgio et al., and Kabir et al.), and we have emphasised that the observed differences may be due to temporal and geographical variability, as well as the use of different methodologies. We have also discussed potential causes of discrepancies, such as the presence of resistant subpopulations or the loss of mobile genetic elements during analysis. We have modified the conclusions by strengthening them and highlighting the importance of a One Health approach, which integrates veterinary and human surveillance, to address the zoonotic risk and the need for coordinated monitoring systems. We emphasised that the combined presence of virulence genes and multiple resistances reinforces the hypothesis of SEZ's increasing pathogenic potential, comparable to that of other major clinically relevant streptococci.
The requested changes to the figures and text lines have been made, as requested.

Reviewer 3 Report
Comments and Suggestions for Authors
Dear authors
Please, find below some suggestions.
Introduction: Streptococcus equi subsp. zooepidemicus (SEZ) has been implicated in several significant outbreaks and epidemics in the past. To broaden the scope of references cited in the first paragraph of the Introduction (and how extensively SEZ is affecting animal species), additional examples could be included, such as outbreaks in dogs associated with haemorrhagic pneumonia, and the extensive epidemic in the Icelandic horse population that affected approximately 80,000 horses.
- 1) Dog. Mangano ER, Jones GMC, Suarez-Bonnet A, Waller AS, Priestnall SL. Streptococcus zooepidemicus in dogs: Exploring a canine pathogen through multilocus sequence typing. Vet Microbiol. 2024 May;292:110059. doi: 10.1016/j.vetmic.2024.110059. Epub 2024 Mar 24. PMID: 38554599.
- 2) Björnsdóttir S, Harris SR, Svansson V, Gunnarsson E, Sigurðardóttir ÓG, Gammeljord K, Steward KF, Newton JR, Robinson C, Charbonneau ARL, Parkhill J, Holden MTG, Waller AS. Genomic Dissection of an Icelandic Epidemic of Respiratory Disease in Horses and Associated Zoonotic Cases. mBio. 2017 Aug 1;8(4):e00826-17. doi: 10.1128/mBio.00826-17. PMID: 28765219; PMCID: PMC5539424.
It may be beneficial to include a few sentences in the Introduction to outline the extend of AMR in SEZ within the horse population, particularly data from other regions of Italy or neighbouring countries.
Briefly reporting the size of the horse population in Italy and in the regions targeted by the study, would help highlighting the close interaction between humans and horses (to support the One health perspective discussed in lines 58-64). A relevant example of the horse/human one health dynamic is West Nile Fever disease and the role of equine veterinarians in the surveillance of this vector born viral disease affecting both humans and equidaes. WNV is quite prominent at the moment and this could be an example to consider (even if not bacterial by nature) to strengthen your statement.
Nasal and genital swabs have been used in this study. SEZ is primary isolated from respiratory and reproductive tracts in horses, which explains the type of sample used for your study. This need to be explained, with supporting data and references (e.g. SEZ is the pathogen most commonly isolated from the uterus of mares … ).
Results:
3.2.: You could improve consistency in the formulation of antibiotics used. For example, full names are used in lines 135 to 141, abbreviations are used in lines 144 to 152. I would also suggest to define the abbreviations in Figure 2 legend.
Figure 1 reports the overall results of antimicrobial susceptibility. Could you further define. For example, could you provide sub-results for nasopharyngeal strains and genital swabs. Is there a significant association between the origin and Susceptibility/Resistance? Is there significant association between species (Horse vv Donkey) and Susceptibility/Resistance?
Figure 2: would it be more interesting to report this as a Table, with more details (e.g. % in horses and donkeys).
Table 2: needs to be reformatted, the right column(s) are only partially visible.
3.3: SEZ superantigens. You report the presence of speI and speK. These are the nomenclature used for S. pyogenes superantigens. The SEZ equivalent have been investigated and described. It would be important to use the SEZ (and Streptococcus equi spp equi) nomenclature for these sAgs alongside the S. pyogenes names in brackets (i.e. speI is seeI; speK is speS a recently described sAg, cf following ref Dominguez-Medina CC, Rash NL, Robillard S, Robinson C, Efstratiou A, Broughton K, Parkhill J, Holden MTG, Lopez-Alvarez MR, Paillot R, Waller AS. SpeS: A Novel Superantigen and Its Potential as a Vaccine Adjuvant against Strangles. Int J Mol Sci. 2020 Jun 23;21(12):4467. doi: 10.3390/ijms21124467. PMID: 32586031; PMCID: PMC7352279.).
Discussion:
It would be beneficial to provide some information about the use of antibiotics/antimicrobials in the Italian horse population (type, purpose, frequency, evolution in time etc.).
Lines 199 to 203: seeI (speI) and speS (speK) activities in equids have been described so references could be used to support statement (… capable stimulate the immune system to release inflammatory cytokines, …).
Line 209: Nocera et al (2023).
Lines 268-271: comparison is made with some of the information reviewed by Kabir et al. Special mention is made of resistance rate in the US. It would be more useful to compare rates with other European countries. Data are available from the UK and France. While Kabir et al is a review, I would suggest looking at the specific studies for more accurate details (cf Isgren CM, Williams NJ, Fletcher OD, Timofte D, Newton RJ, Maddox TW, Clegg PD, Pinchbeck GL. Antimicrobial resistance in clinical bacterial isolates from horses in the UK. Equine Vet J. 2022 Mar;54(2):390-414. doi: 10.1111/evj.13437. Epub 2021 May 4. PMID: 33566383. and Léon A, Castagnet S, Maillard K, Paillot R, Giard JC. Evolution of In Vitro Antimicrobial Susceptibility of Equine Clinical Isolates in France between 2016 and 2019. Animals (Basel). 2020 May 7;10(5):812. doi: 10.3390/ani10050812. PMID: 32392891; PMCID: PMC7278474.)
Conclusion: The manuscript suggests that SEZ should be considered an emerging pathogen requiring more systematic and coordinated surveillance. It is important to clarify which species this statement refers to, as SEZ is not considered an emerging pathogen within the equine industry. To strengthen this point, it would be helpful to discuss the extend of SEZ diagnosis and epidemiology in the equine industry (especially in Europe), which could provide valuable context and support for the conclusions drawn.
References:
- The reference Bosica et al, 2023 is listed twice (ref 8 and 19).
Author Response
(C1)Streptococcus equi subsp. zooepidemicus (SEZ) has been implicated in several significant outbreaks and epidemics in the past. To broaden the scope of references cited in the first paragraph of the Introduction (and how extensively SEZ is affecting animal species), additional examples could be included, such as outbreaks in dogs associated with haemorrhagic pneumonia, and the extensive epidemic in the Icelandic horse population that affected approximately 80,000 horses.1) Dog. Mangano ER, Jones GMC, Suarez-Bonnet A, Waller AS, Priestnall SL. Streptococcus zooepidemicus in dogs: Exploring a canine pathogen through multilocus sequence typing. Vet Microbiol. 2024 May;292:110059. doi: 10.1016/j.vetmic.2024.110059. Epub 2024 Mar 24. PMID: 38554599.2) Björnsdóttir S, Harris SR, Svansson V, Gunnarsson E, Sigurðardóttir ÓG, Gammeljord K, Steward KF, Newton JR, Robinson C, Charbonneau ARL, Parkhill J, Holden MTG, Waller AS. Genomic Dissection of an Icelandic Epidemic of Respiratory Disease in Horses and Associated Zoonotic Cases. mBio. 2017 Aug 1;8(4):e00826-17. doi: 10.1128/mBio.00826-17. PMID: 28765219; PMCID: PMC5539424.It may be beneficial to include a few sentences in the Introduction to outline the extend of AMR in SEZ within the horse population, particularly data from other regions of Italy or neighbouring countries.Briefly reporting the size of the horse population in Italy and in the regions targeted by the study, would help highlighting the close interaction between humans and horses (to support the One health perspective discussed in lines 58-64). A relevant example of the horse/human one health dynamic is West Nile Fever disease and the role of equine veterinarians in the surveillance of this vector born viral disease affecting both humans and equidaes. WNV is quite prominent at the moment and this could be an example to consider (even if not bacterial by nature) to strengthen your statement.Nasal and genital swabs have been used in this study. SEZ is primary isolated from respiratory and reproductive tracts in horses, which explains the type of sample used for your study. This need to be explained, with supporting data and references (e.g. SEZ is the pathogen most commonly isolated from the uterus of mares … ).
(R1)We thank the reviewer for the suggestion. In the revised introduction, we have added references to SEZ outbreaks in dogs (Mangano et al., 2024) and the epidemic affecting Icelandic horses (Björnsdóttir et al., 2017), in order to emphasise the broad host range and impact of SEZ across different species of animal. These additions serve to reinforce the notion of SEZ as a multi-host opportunistic pathogen.We have also incorporated additional sentences summarising SEZ resistance rates in other Italian regions and neighbouring countries. Furthermore, we have included data on the equine population in Italy and the Abruzzo and Molise regions to emphasise the frequent interactions between humans and horses. We have clarified that SEZ is primarily isolated from the respiratory and reproductive tracts of horses, justifying the use of nasal and genital swabs. We have added supporting references indicating that SEZ is the most common pathogen isolated from the uterus of mares and the equine respiratory tract.
(C2) 3.2.: You could improve consistency in the formulation of antibiotics used. For example, full names are used in lines 135 to 141, abbreviations are used in lines 144 to 152. I would also suggest to define the abbreviations in Figure 2 legend. Figure 1 reports the overall results of antimicrobial susceptibility. Could you further define. For example, could you provide sub-results for nasopharyngeal strains and genital swabs. Is there a significant association between the origin and Susceptibility/Resistance? Is there significant association between species (Horse vv Donkey) and Susceptibility/Resistance? Figure 2: would it be more interesting to report this as a Table, with more details (e.g. % in horses and donkeys).
(R2)We have revised the text to ensure consistent use of antibiotic names: full names are given at first mention, followed by abbreviations in parentheses. Abbreviations are now defined in the legend of Figure 2 for clarity. The data on the various associations of antibiotic resistance results for different types of swab were not processed because there are far fewer urogenital swabs than nasal swabs, which could lead to a misleading assessment of significance. Nevertheless, we deemed it appropriate to evaluate the data, and plan to do so again once comparable numbers of samples of different types are available.
(C3)SEZ superantigens. You report the presence of speI and speK. These are the nomenclature used for S. pyogenes superantigens. The SEZ equivalent have been investigated and described. It would be important to use the SEZ (and Streptococcus equi spp equi) nomenclature for these sAgs alongside the S. pyogenes names in brackets (i.e. speI is seeI; speK is speS a recently described sAg, cf following ref Dominguez-Medina CC, Rash NL, Robillard S, Robinson C, Efstratiou A, Broughton K, Parkhill J, Holden MTG, Lopez-Alvarez MR, Paillot R, Waller AS. SpeS: A Novel Superantigen and Its Potential as a Vaccine Adjuvant against Strangles. Int J Mol Sci. 2020 Jun 23;21(12):4467. doi: 10.3390/ijms21124467. PMID: 32586031; PMCID: PMC7352279.).
(R3)We have chosen to use the unified nomenclature proposed by Commons et al. (2014), as indicated in the Dominguez-Medina et al. (2020) article you cited.
(C4) It would be beneficial to provide some information about the use of antibiotics/antimicrobials in the Italian horse population (type, purpose, frequency, evolution in time etc.). Lines 199 to 203: seeI (speI) and speS (speK) activities in equids have been described so references could be used to support statement (… capable stimulate the immune system to release inflammatory cytokines, …).Line 209: Nocera et al (2023). Lines 268-271: comparison is made with some of the information reviewed by Kabir et al. Special mention is made of resistance rate in the US. It would be more useful to compare rates with other European countries. Data are available from the UK and France. While Kabir et al is a review, I would suggest looking at the specific studies for more accurate details (cf Isgren CM, Williams NJ, Fletcher OD, Timofte D, Newton RJ, Maddox TW, Clegg PD, Pinchbeck GL. Antimicrobial resistance in clinical bacterial isolates from horses in the UK. Equine Vet J. 2022 Mar;54(2):390-414. doi: 10.1111/evj.13437. Epub 2021 May 4. PMID: 33566383. and Léon A, Castagnet S, Maillard K, Paillot R, Giard JC. Evolution of In Vitro Antimicrobial Susceptibility of Equine Clinical Isolates in France between 2016 and 2019. Animals (Basel). 2020 May 7;10(5):812. doi: 10.3390/ani10050812. PMID: 32392891; PMCID: PMC7278474.)
(R4)Thank you for this valuable suggestion. We have added a paragraph in the Introduction providing background information on antimicrobial use in Italian horses, including the main classes prescribed. While we initially compared our findings with resistance data reported in the US, we now expanded the Discussion by including European studies for a more relevant comparison as suggested. Some changes have been made to the text regarding superantigens.
(C5)The manuscript suggests that SEZ should be considered an emerging pathogen requiring more systematic and coordinated surveillance. It is important to clarify which species this statement refers to, as SEZ is not considered an emerging pathogen within the equine industry. To strengthen this point, it would be helpful to discuss the extend of SEZ diagnosis and epidemiology in the equine industry (especially in Europe), which could provide valuable context and support for the conclusions drawn.
(R5) The conclusions have been amended as requested.
(C6)The reference Bosica et al, 2023 is listed twice (ref 8 and 19).
(R6) Corrected.

Round 2
Reviewer 3 Report
Comments and Suggestions for Authors
Dear Authors
Thank you for your revised manuscript.
References 8 and 32 may need to be reviewed (formatting issue, first name not abbreviated and first name abbreviation in front of the surname, respectively).